# Certification of Distributional Individual Fairness

**Matthew Wicker**
The Alan Turing Institute
mwicker@turing.ac.uk

**Vihari Piratia**
University of Cambridge
vp421@cam.ac.uk

**Adrian Weller**
University of Cambridge &
The Alan Turing Institute

## Abstract

Providing formal guarantees of algorithmic fairness is of paramount importance to socially responsible deployment of machine learning algorithms. In this work, we study formal guarantees, i.e., certificates, for individual fairness (IF) of neural networks. We start by introducing a novel convex approximation of IF constraints that exponentially decreases the computational cost of providing formal guarantees of local individual fairness. We highlight that prior methods are constrained by their focus on global IF certification and can therefore only scale to models with a few dozen hidden neurons, thus limiting their practical impact. We propose to certify *distributional* individual fairness which ensures that for a given empirical distribution and all distributions within a $\gamma$-Wasserstein ball, the neural network has guaranteed individually fair predictions. Leveraging developments in quasi-convex optimization, we provide novel and efficient certified bounds on distributional individual fairness and show that our method allows us to certify and regularize neural networks that are several orders of magnitude larger than those considered by prior works. Moreover, we study real-world distribution shifts and find our bounds to be a scalable, practical, and sound source of IF guarantees.

## 1 Introduction

There is a growing concern about the potential of machine learning models to perpetuate and amplify discrimination (Barocas & Selbst, 2016). Machine learning algorithms have been put forward to automate decision making in a variety of fairness-sensitive domains, such as healthcare (Davenport & Kalakota, 2019), employment (Ding et al., 2021), and criminal justice (Dressel & Farid, 2018; Zilka et al., 2022). It has been demonstrated that such models produce biased outcomes that unfairly disadvantage certain individuals or groups (Seyyed-Kalantari et al., 2021; Yurochkin & Sun, 2020). To combat this, there has been a surge of interest in algorithmic fairness metrics (Mehrabi et al., 2021). Fairness metrics provide practitioners with a means to quantify the extent of bias within a model and facilitate development of heuristics for debiasing models (Madras et al., 2018; Yurochkin et al., 2019). However, relying solely on heuristic debiasing methods may not be satisfactory for justifying the deployment of machine learning models as they do not provide guarantees of fairness. To address this concern, there has been a recent focus on certification approaches, which provide formal guarantees that a model adheres to a specified fairness criterion (John et al., 2020; Benussi et al., 2022; Khedr & Shoukry, 2022). Providing certified guarantees of fairness is of utmost importance as it offers users, stakeholders, and regulators formal assurances that the predictions of a model align with a rigorous standards of fairness. These guarantees serve as a powerful tool to promote trust, accountability, and ethical deployment of machine learning models.

In this work, we study the problem of certifying *individual fairness* (IF) in neural networks (NNs). Individual fairness enforces the intuitive property that a given neural network issues similar predictions for all pairs of *similar* individuals (Dwork et al., 2012). Individuals are considered similar if they differ only with respect to protected attributes (e.g., race, gender, or age) or features correlated with the protected attributes. This similarity is captured by a fair distance metric which can be learned by

37th Conference on Neural Information Processing Systems (NeurIPS 2023).

querying human experts or extracted from observed data (Mukherjee et al., 2020). Given a NN and a fairness metric, recent works have established procedures for certifying that a NN conforms to a given IF definition, which is a crucial step in developing and deploying fair models (Benussi et al., 2022; Khedr & Shoukry, 2022). While current approaches to IF guarantees for NNs are effective for simple problems, they face scalability issues when dealing with NNs containing more than a few dozen neurons. This limitation is due to their emphasis on global individual fairness, which guarantees the IF property for every input within the NN's domain (Benussi et al., 2022; Khedr & Shoukry, 2022). Although global IF certification is the gold standard, it places constraints on extreme outliers, such as ensuring loan decisions are fair to a three-year-old billionaire applying for a small business loan. Subsequently, as the dimension of the input space grows, solving an optimization problem that covers the entire domain becomes either impossible for current solvers to deal with or computationally prohibitive. On the other hand, distributional individual fairness (DIF), first proposed in Yurochkin et al. (2019), enforces that a model's predictions are individually fair w.r.t. a family of distributions that are within a $\gamma-$Wasserstein ball of the empirical distribution over observed individuals. The focus on distributions removes the constraints on unrealistic individuals while enforcing fairness on the relevant part of the input domain. As such, providing certification of DIF can provide strong guarantees of fairness that scale to even relatively large neural network models. However, prior works in DIF focus only on heuristic debiasing techniques (Yurochkin et al., 2019).

In this work, we provide the first formal certificates of DIF. As it cannot be computed exactly, we present a framework for bounding distributional individual fairness. We start by presenting a novel convex relaxation of local IF constraints, i.e., IF with respect to a single prediction. Our approach to local IF certification offers an exponential computational speed-up compared with prior approaches. Building on this efficient bound, we are able to propose an optimization problem whose solution, attainable due to the problem's quasi-convexity, produces sound certificates of DIF. Further, our certified bounds can be easily incorporated as a learning objective, enabling efficient individual fairness training. In a series of experiments, we establish that our method is able to certify that IF holds for meaningful, real-world distribution shifts. Further, our proposed regularization is able to certifiably fair neural networks that are two orders of magnitude larger than those considered in previous works in a fraction of the time. We highlight the following key contributions of this work:

- We prove a convex relaxation of local individual fairness, enabling exponentially more efficient local IF certification compared with prior mixed integer linear programming approaches.

- We formulate a novel optimization approach to certificates of local and distribution individual fairness and leverage quasi-convex optimization to provide the first formal upper and lower bounds, i.e., certificates, on distributional individual fairness.

- We empirically demonstrate the tightness of our distributional individual fairness guarantees compared with real-world distributional shifts, and demonstrate the considerable scalability benefits of our proposed method.

## 2 Related Work

Individual fairness, originally investigated by Dwork et al. (2012), establishes a powerful and intuitive notion of fairness. Subsequent works have focused on defining the individual fairness metrics (Mukherjee et al., 2020; Yurochkin & Sun, 2020), expanding the definition to new domains (Gupta & Kamble, 2021; Xu et al.; Doherty et al., 2023), and importantly, guaranteeing that a model conforms to individual fairness (John et al., 2020; Ruoss et al., 2020; Benussi et al., 2022; Khedr & Shoukry, 2022; Peychev et al., 2022). In (John et al., 2020) the authors consider a relaxation of individual fairness and present methods to verify that it holds for linear classifiers. In (Yeom & Fredrikson, 2020; Peychev et al., 2022) the authors extend randomized smoothing to present statistical guarantees on individual fairness for neural networks. These guarantees are much weaker than the sound guarantees presented in Benussi et al. (2022); Khedr & Shoukry (2022) which are based on mixed integer linear programming (MILP). Both of these methods focus solely on the global notion of individual fairness which proves to only scale to neural networks with a few dozen hidden neurons. Our work relaxes the need to certify a property globally allowing us to scale to neural networks that are orders of magnitude larger than what can be considered by prior works. Further, prior works have sought to debias models according an individual fairness criterion (Yurochkin et al., 2019; Benussi et al., 2022; Khedr & Shoukry, 2022). In Yurochkin et al. (2019) the authors present a method for debiased training relying on PGD which is insufficient for training a NN with strong fairness guarantees (Benussi et al., 2022).

In (Benussi et al., 2022; Khedr & Shoukry, 2022) the authors use linear programming formulations of fairness during training which leads to strong IF guarantees but at a large computational cost and greatly limited scalability.

Distributional individual fairness was originally proposed by Yurochkin et al. (2019) and the importance of distributional robustness of notions of fairness has been underscored in several recent woks (Sagawa et al., 2019; Sharifi-Malvajerdi et al., 2019; Taskesen et al., 2020; Mukherjee et al., 2022). To the best of our knowledge, this work is the first that provides formal certificates that guarantee a model satisfies distributional individual fairness. We provide further discussion of works related to distributional robustness in Appendix C.

## 3 Background

We consider a supervised learning scenario in which we have a set of $n$ feature-label pairs drawn from an unknown joint distribution $\{(x^{(i)}, y^{(i)})\}_{i=1}^n \sim P^0(x, y)$ with $x \in \mathbb{R}^m$ and $y \in \mathbb{R}^k$. We consider a feed forward neural network (NN) as a function $f^\theta : \mathbb{R}^m \to \mathbb{R}^k$, parametrised by a vector $\theta \in \mathbb{R}^p$. We define a local version of the individual fairness definition that is studied in (John et al., 2020; Benussi et al., 2022) to evaluate algorithmic bias:

**Definition 1.** *Local $\delta$-$\epsilon$-Individual Fairness* Given $\delta > 0$ and $\epsilon \geq 0$, we say that $f^\theta$ is locally individually fair (IF) with respect to fair distance $d_{fair}$ and input $x'$ iff:

$$\forall x'' \ s.t. \ d_{fair}(x', x'') \leq \delta \implies |f^\theta(x') - f^\theta(x'')| \leq \epsilon$$

For a given value of $\delta$ we use the notation $\mathcal{I}(f^\theta, x, \delta)$ to denote the function that returns the largest value of $\epsilon$ such that the local IF property holds. Further, $\mathcal{I}(f^\theta, x, \delta) = 0$ holds if a network is perfectly locally individually fair. Throughout the paper, we will refer to the value $\epsilon$ as the IF *violation* as this is the amount by which the perfect fairness property is violated.

Several methods have been investigated for crafting fair distance metrics (Mukherjee et al., 2020). Methods in this work can flexibly handle a majority of the currently proposed metrics. We discuss the particulars in Appendix D. While certifying local IF is sufficient for proving that a given prediction is fair w.r.t. a specific individual, it does not suffice as proof that the model will be fair for unseen individuals at deployment time. To this end, prior works have investigated a global individual fairness definition. Global $\epsilon$-$\delta$ individual fairness holds if and only if $\forall x \in \mathbb{R}^m, \mathcal{I}(f^\theta, x, \delta) \leq \epsilon$. However, global individual fairness is a strict constraint and training neural networks to satisfy global individual fairness comes at a potentially prohibitive computational cost. For example, training a neural network with only 32 hidden units can take up to 12 hours on a multi-GPU machine (Benussi et al., 2022).

## 4 Distributional Individual Fairness

In this section, we describe *distributional individual fairness* (DIF) as a notion that can ensure that neural networks are individually fair at deployment time while not incurring the considerable overhead of global training and certification methods. DIF ensures that individuals in the support of $P^0$ and distributions close to $P^0$ are treated fairly. To measure the distributions close to $P^0$ we consider the $p$-Wasserstein distance between distributions, $W_p(P^0, \cdot)$. Further, we denote the ball of all distributions within $\gamma$ $p$-Wasserstein distance of $P^0$ as $\mathcal{B}_{\gamma,p}(P^0)$. To abbreviate notation, we drop the $p$ from the Wasserstein distance, noting that any integer $p \geq 1$ can be chosen.

**Definition 2.** *Distributional $\gamma$-$\delta$-$\epsilon$-Individual Fairness* Given individual fairness parameters $\delta > 0$ and $\epsilon \geq 0$, and distributional parameter $\gamma > 0$, we say that $f^\theta$ is distributionally individually fair (DIF) with respect to fair distance $d_{fair}$ if and only if the following condition holds:

$$\sup_{P^\star \in \mathcal{B}_\gamma(P^0)} \left( \mathbb{E}_{x \sim P^\star}[\mathcal{I}(f^\theta, x, \delta)] \right) \leq \epsilon \tag{1}$$

Intuitively, distributional individual fairness ensures that for all distributions in a Wasserstein ball around $P^0$ we maintain individual fairness on average. In Figure 1, we visualize the difference between local, distributional, and global IF in a toy two dimensional settings. Prior work that focuses on DIF only provides a heuristic method based on the PGD attack to train debiased NNs and therefore

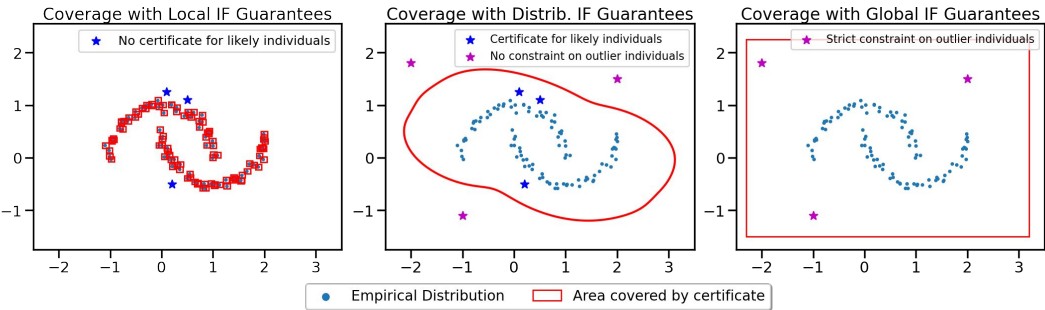

Figure 1: Using the halfmoons dataset we illustrate the benefits of DIF. **Left:** local IF does not do enough to provide guarantees on individuals we are likely to see. **Right:** Global IF provides a strict constraint on anomalous points, leading to scalability issues. **Center**: Distributional IF covers likely individuals without over-constraining the problem leading to scalable and reliable fairness guarantees.

provide no formal guarantees (Yurochkin et al., 2019). Below we state two key computational problems in the study of guarantees for distributional individual fairness:

**Problem 1.** *Certification of Distributional Individual Fairness: Given a neural network, $f^\theta$ and parameters $\delta$ and $\gamma$, compute a value $\bar{\epsilon}$ such that $\bar{\epsilon}$ provably upper-bounds the left hand side of Equation (1). An optimal certificate is the smallest such $\bar{\epsilon}$ such that Equation (1) holds.*

The problem of certifying distributional individual fairness is critical to ensuring that the definition can be used in practice as it allows for regulators and auditors to have similar assurances to global individual fairness, but only on what is deemed to be the relevant portion of the input space as controlled by $\gamma$.

**Problem 2.** *Distributionally Individually Fair Training: Given a randomly initialized neural network, $f^\theta$ and parameters $\delta$ and $\gamma$ and a dataset drawn from $P^0$, learn a neural network that approximately minimizes both the standard loss, $\mathcal{L}$, and the DIF violation:*

$$\underset{\theta \in \Theta}{\arg\min} \, \mathbb{E}_{P^0}\left[\mathcal{L}(f^\theta, x, y)\right] + \sup_{P^\star \in \mathcal{B}_{\gamma, p}(P^0)} \left(\mathbb{E}_{P^\star}[\mathcal{I}(f^\theta, x, \delta)]\right) \tag{2}$$

Problem 2 is tantamount to certified debiased training of a neural network w.r.t the DIF definition. Certified debiasing places a focus on formal guarantees that is not present in prior distributional individual fairness work.

## 5 Methodology

To address Problems 1 and 2 we start by providing a novel and efficient computation for certificates of local individual fairness. We then build on this efficient guarantee to compute formal bounds and proposed regularizers for certified DIF training.

### 5.1 Certifying Local Individual Fairness

A key step in certification of distributional individual fairness is proving that a given neural network satisfies local individual fairness i.e., Definition 1. While there are many methods for efficiently certifying local robustness properties, individual fairness must be certified w.r.t. a ball of inputs whose shape and size is determined by $d_{\text{fair}}$. Existing works in crafting individual fairness metrics use Mahalanobis metrics to express the fair distance between two individuals, i.e., $d_{\text{fair}}(x, x') = \sqrt{(x - y)^\top S^{-1}(x - y)}$ where $S$ is a positive semi-definite matrix $\in \mathbb{R}^{m \times m}$. In order to certify this more expressive class of specifications prior works use linear bounding techniques as part of a larger MILP formulation (Benussi et al., 2022). In this work, we show that one can in fact use exponentially more efficient bound propagation techniques developed for robustness certification to certify local individual fairness. In order to do so, we provide the following bound that over-approximates the $d_{\text{fair}}$ metric ball with an orthotope:

**Theorem 5.1.** *Given a positive semi-definite matrix $S \in \mathbb{R}^{m \times m}$, a feature vector $x' \in \mathbb{R}^m$, and a similarity theshold $\delta$, all vectors $x'' \in \mathbb{R}^m$ satisfying $d_S(x', x'') \leq \delta$ are contained within the axis-aligned orthope:*

$$\left[ x' - \delta\sqrt{d}, x' + \delta\sqrt{d} \right]$$

*where $d = diag(S)$, the vector containing the elements along the diagonal of $S$.*

Theorem 5.1 holds as well for weighted $\ell_p$ fairness metrics by setting the diagonal entries of $S$ to the weights for metric provides an over approximation of the $\ell_\infty$ metric, which is also a valid over-approximation for any $p \leq \infty$. The key consequence of Theorem 5.1 is that it directly enables efficient certification of local $\delta$-$\epsilon$ individual fairness. By over-approximating the fair distance with an axis-aligned orthotope, we are able to leverage the efficient methods developed for robustness certification (Gowal et al., 2018). Such methods allow us to take the given input interval and compute output bounds $[y^L, y^U]$ such that $\forall x \in [x' - \delta\sqrt{d}, x' + \delta\sqrt{d}], f^\theta(x) \in [y^L, y^U]$. Subsequently, it is clear to see that if $|y^U - y^L| \leq \epsilon$ then we have certified that $\forall x' \in [x - \delta\sqrt{d}, x + \delta\sqrt{d}] \implies |f^\theta(x) - f^\theta(x')| \leq \epsilon$ which proves that local individual fairness holds according to Definition 1. Computing such output bounds requires the computational complexity of two forward Mirman et al. (2018); Gowal et al. (2018). MILP approaches, on the other hand, requires complexity cubic in the largest number of neurons in a layer plus requires exponentially (in the number of problem variables) many MILP iterations Benussi et al. (2022). We denote the function that takes the interval corresponding to an individual $x$ and value $\delta$ and returns $|y^U - y^L|$ with the notation $\overline{\mathcal{I}(f^\theta, x, \delta)}$. Given that bound propagation methods are sound, this value over-approximates the $\epsilon$ function in Definition 1. We emphasize that this approach to local fairness certification only takes two forward passes through the neural network, and can therefore be used as real-time decision support. We provide exact details of the computations of this upper bound in Appendix D. By exploring the orthotope using methods developed for adversarial attacks, e.g., PGD (Madry et al., 2017), we can find an input $x^\star$ that approximately maximizes $|f^\theta(x') - f^\theta(x^\star)|$. Subsequently, this value is a valid lower-bound on the local IF violation around $x$. As such, we will denote the value produced by PGD as $\underline{\mathcal{I}(f^\theta, x, \delta)}$.

### 5.2 Certifying Distributional Individual Fairness

In this section we leverage the efficient upper bound on $\mathcal{I}(f^\theta, x, \delta)$ to prove bounds on distributional individual fairness. We first highlight that computing the Wasserstein distance between two known distributions is non-trivial, and that in general we do not have access to the joint distribution $P^0$. Thus, computing the Wasserstein ball around the data generating distribution is infeasible. To overcome this, as is traditionally done in the distributional robustness literature, we make the practical choice of certifying distributional individual fairness w.r.t. the empirical distribution $\hat{P}^0$ which is the probability mass function with uniform density over a set of $n$ observed individuals (Sinha et al., 2017). While this introduces approximation in the finite-sample regime, we can bound the absolute difference between our estimate and the true value by using Chernoff's inequality, discussion in Appendix D. Given the empirical distribution of individuals as well as IF parameters $\gamma$ and $\delta$, we pose an optimization objective whose solution is the tightest value of $\epsilon$ satisfying Definition 2.

$$\epsilon^* = \max_{\phi^{(1..n)}} \frac{1}{n} \sum_{i=1}^n \mathcal{I}(f^\theta, x^{(i)} + \phi^{(i)}, \delta), \quad s.t., \quad \phi^{(i)} \in \mathbb{R}^m, \frac{1}{n} \sum_{i=1}^n ||\phi^{(i)}||^p \leq \gamma^p \quad (3)$$

Unfortunately, computing the optimal values of $\phi^{(i)}$ even when $n = 1$ is a non-convex optimization problem which is known to be NP-hard (Katz et al., 2017). We restate this optimization problem in terms of our bounds on $\mathcal{I}(f^\theta, x, \delta)$ to get formal lower ($\underline{\epsilon}$) and upper ($\overline{\epsilon}$) bounds on $\epsilon^\star$ such that $\underline{\epsilon} \leq \epsilon^\star \leq \overline{\epsilon}$.

### 5.3 Bounding Distributional Individual Fairness

**Lower-bound on DIF** Observe that any selection of $\{\phi^{(i)}\}_{i=1}^n$ satisfying the constraints represents a feasible solution to the optimization problem and therefore a lower-bound on the maximization problem of interest. Thus, the lower-bound can be stated as:

$$\underline{\epsilon} = \max_{\phi^{(1..n)}} \frac{1}{n} \sum_{i=1}^n \underline{\mathcal{I}(f^\theta, x^{(i)} + \phi^{(i)}, \delta)} \quad s.t., \quad \phi^{(i)} \in \mathbb{R}^m, \frac{1}{n} \sum_{i=1}^n ||\phi^{(i)}||^p \leq \gamma^p \quad (4)$$

We can optimize this lower bound to be tight by observing that for any given $i$, the function $\underline{\mathcal{I}(f^\theta, x^{(i)} + \phi^{(i)}, \delta)}$ is differentiable w.r.t. $\phi^{(i)}$ and is therefore amenable to first-order optimization. In particular, we denote $\phi_0^{(i)}$ to be a randomly selected, feasible assignment of $\phi_0^{(i)}$, we can then gradient ascent to find a locally optimal assignment of $\phi^{(i)}$:

$$\phi_{j+1}^{(i)} \leftarrow \phi_j^{(i)} + \alpha \nabla_{\phi^{(i)}} \underline{\mathcal{I}(f^\theta, x^{(i)} + \phi^{(i)}, \delta)}$$

where $\alpha$ is a learning rate parameter. Subsequently, one could use a projection step in order to ensure that the constraint in Equation (3) is never violated. We note that this is a strict lower-bound unless the Equation (4) is solved exactly and $\underline{\mathcal{I}(f^\theta, x^{(i)} + \phi^{(i)}, \delta)} = \mathcal{I}(f^\theta, x^{(i)} + \phi^{(i)}, \delta)$ for all $i$. Finally, we denote the computed solution to Equation (4) as $\underline{I_\gamma(f^\theta, X, \delta)}$ where $\gamma$ is the distributional parameter of the DIF specification and $X$ is the tensor containing the $n$ individual feature vectors.

**Upper-bound on DIF**  An upper-bound on the maximization in Equation (3), i.e., $\overline{\epsilon} \geq \epsilon^\star$, constitutes a certificate that the model provably satisfies distributional individual fairness for all $\epsilon$ greater than $\overline{\epsilon}$. Given that the optimization posed by Equation (3) is highly non-convex, searching for an optimal solution cannot be guaranteed to converge globally without knowledge of global Lipschitz constants, which are difficult to compute efficiently for large NNs (Fazlyab et al., 2019). Instead, we transform the optimization problem from one over the input space to the space of over-approximate local IF violations:

$$\overline{\epsilon} = \max_{\varphi^{(1..n)}} \quad \frac{1}{n} \sum_{i=1}^n \overline{\mathcal{I}(f^\theta, x, \delta + \varphi^{(i)})} \quad s.t. \quad \varphi^{(i)} \in \mathbb{R}^+, \frac{1}{n} \sum_{i=1}^n ||\varphi^{(i)}||^p \leq \gamma^p \tag{5}$$

The key change to the formulation is that we are no longer considering input transformations parameterized by $\phi^{(i)}$, but instead consider an over-approximation of all possible $\phi^{(i)}$ that are within a $\varphi^{(i)}$-radius around the point $x^{(i)}$. Moreover, it is clear that picking any assignment of the $\varphi^{(i)}$ values will *not* constitute an upper-bound on the DIF violation. However, the global maximizing assignment of $\varphi^{(i)}$ values *is* guaranteed to over-approximate the value of Equation (3), formally stated in Theorem 5.2.

**Theorem 5.2.** *Given an optimal assignment of $\{\varphi^{(i)}\}_{i=1}^n$ in Equation (5), the corresponding $\overline{\epsilon}$ is a sound upper-bound on the DIF violation of the model and therefore, is a certificate that no $\gamma$-Wasserstien distribution shift can cause the individual fairness of the model to exceed $\overline{\epsilon}$.*

The proof of Theorem 5.2 is contained in Appendix E.3. Given that the function $\overline{\mathcal{I}(f^\theta, x, \delta + \varphi^{(i)})}$ is a continuous and monotonically increasing function, we have that the maximization problem in Equation (5) is quasi-convex. After proving this function is also Hölder continuous, one can guarantee convergence in finite time to the optimal solution with bounded error using recent advancements in quasi-convex optimization (Hu et al., 2020; Agrawal & Boyd, 2020; Hu et al., 2022). As the global optimal solution is guaranteed to be an upper-bound, we can use the methods for quasi-convex optimization to compute a certificate for the DIF property. In the Appendix we prove the Hölder continuity of $\mathcal{I}$ and conduct numerical convergence experiments to validate this theory. Given that the result of this optimization is a certified upper bound on the DIF value, we denote its result with $\overline{I_\gamma(f^\theta, X, \delta)}$ where $\gamma$ is the distributional parameter of the DIF formulation. While $\overline{\epsilon}$ is a valid upper-bound on the DIF violation for the given dataset, in the low-data regime one may have non-negligible error due to poor approximation of the data distribution. The finite sample approximation error can be bounded by an application of Hoeffding's inequality as we state in Lemma 1.

**Lemma 1.** *Given an upper-bound, $\overline{\epsilon}$ (a global maximizing assignment to Equation (5)) we can bound the error induced by using a finite sample from above by $\tau$. That is, the estimate $\overline{\epsilon}$ is within $\tau$ of the true expectation of $\overline{\epsilon}$ with probability $1 - \lambda$ as long as $n$ is at least $(-1/2\tau^2) \log(\lambda/2)$*

### 5.4  Training Distributionally Individually Fair NNs

It is well-known that guarantees for certification agnostic neural networks can be vacuous (Gowal et al., 2018; Wicker et al., 2021). To combat this, strong empirical results have been obtained for neural networks trained with certified regularizers (Benussi et al., 2022; Khedr & Shoukry, 2022). In this work, we have proposed three novel, differentiable bounds on certified individual fairness. We

start by proposing a loss on certified local individual fairness similar to what is proposed in Benussi et al. (2022) but substituted with our significantly more efficient interval bound procedure:

$$\mathcal{L}_{\text{F-IBP}} = \mathcal{L}(f^\theta, X, Y) + \alpha \overline{\mathcal{I}_0(f^\theta, X, \delta)}$$

where $\alpha$ is a weighting that trades off the fair loss with the standard loss. When $\gamma = 0$, we recover exactly a local constraint on individual fairness for each $x^{(i)} \in X$. By increasing the $\gamma$ from zero we move from a local individual fairness constraint to a DIF constraint:

$$\mathcal{L}_{\text{U-DIF}} = \mathcal{L}(f^\theta, X, Y) + \alpha \overline{\mathcal{I}_\gamma(f^\theta, X, \delta)}$$

where U-DIF stands for upper-bound on DIF. This loss comes at the cost of increased computational time, but is a considerably stronger regularizer. In fact, given that the upper bound on certified DIF might be vacuous, and therefore difficult to optimize, at the start of training, we also propose a training loss that optimizes the lower bound on DIF as it may be empirically easier to optimize and serves as a middle ground between F-IBP and U-DIF:

$$\mathcal{L}_{\text{L-DIF}} = \mathcal{L}(f^\theta, X, Y) + \alpha \underline{\mathcal{I}_\gamma(f^\theta, X, \delta)}$$

## 6 Experiments

In this section we empirically validate our proposed method on a variety of datasets. We first describe the datasets and metrics used. We then cover a wide range of experimental ablations and validate our bounds on real-world distribution shifts. We conclude the section with a study of how IF training impacts other notions of fairness.[1]

**Datasets** We benchmark against the Adult, Credit, and German datasets from the UCI dataset repository (Dua & Graff, 2017). The German or Satlog dataset predicts credit risk of individuals. The Credit dataset predicts if an individual will default on their credit. The Adult dataset predicts if an individuals income is greater than 50 thousand dollars. We additionally consider three datasets from the Folktables datasets, Income, Employ and Coverage, which are made up of millions of data points curated from the 2015 to 2021 US census data (Ding et al., 2021). The Income dataset predicts if an individual makes more than 50 thousand US dollars, the Employ dataset predicts if an individual is employed, and the Coverage dataset predicts if an individual has health insurance. For each dataset gender is considered the protected attribute.

**Metrics** We consider four key metrics. For performance, we measure accuracy which is computed over the entire test-set. Next we consider the local fairness certificates (LFC), which is taken as $1/n \sum_{i=1}^n \overline{\mathcal{I}_0(f^\theta, x_i, \delta)}$. Next we consider the empirical distributional fairness certificate (E-DFC) which is the maximum of the LFC taken over a set of observed distribution shifts. That is, we observe a set of individuals drawn from a distribution shift, $x^{(k)} \sim P^k$, where $P^k$ represents a shift of geographic context (e.g., $P^0$ being individuals from California and $P^1$ being individuals from North Carolina) or a shift in time. E-LFC is then defined as $\max_{j \in [k]} 1/n \sum_{i=0}^n \overline{\mathcal{I}_0(f^\theta, x_i^{(j)}, \delta)}$. Finally, we consider the adversarial distributional fairness certification (A-DFC) which is our computed value of $\overline{\mathcal{I}_\gamma(f^\theta, x_i, \delta)}$. Unless stated otherwise, we use $\delta = 0.05$, $\gamma = 0.1$, and with respect to 1000 test-set individuals. Complete experimental details are given in Appendix A.

### 6.1 Comparing Training Methods

In Table 1, we compare different individual fairness training methods on three datasets from Ding et al. (2021). For each dataset, we train a two layer neural network with 256 hidden neurons per layer. We compare with fairness through unawareness (FTU) which simply removes the sensitive feature and sensitive subspace robustness (SenSR) (Yurochkin et al., 2019). We observe a consistent decrease in accuracy as we increase the strictness of our constraint. While FTU has the best performance, we are unable to compute strong IF guarantees. This result is expected given that it is well-known that guaranteeing the performance of certification-agnostic networks is empirically challenging (Gowal et al., 2018). Similarly, we observe that SenSR has a positive effect on the DIF guarantee while paying a 2-5% accuracy cost. When only enforcing local guarantees with F-IBP, we observe a 3-5%

---

[1]Code to reproduce experiments can be found at: `https://github.com/matthewwicker/DistributionalIndividualFairness`

|        | Income | | | | Employ | | | | Coverage | | | |
|--------|-------|-------|-------|-------|-------|-------|-------|-------|-------|-------|-------|-------|
|        | Acc   | LFC   | E-DFC | A-DFC | Acc   | LFC   | E-DFC | A-DFC | Acc   | LFC   | E-DFC | A-DFC |
| FTU    | **0.820** | 0.999 | 0.999 | 1.000 | **0.809** | 0.891 | 0.906 | 0.999 | **0.721** | 1.000 | 1.000 | 1.000 |
| SenSR  | 0.782 | 0.895 | 0.931 | 0.995 | 0.743 | 0.247 | 0.300 | 0.639 | 0.709 | 0.538 | 0.865 | 0.899 |
| F-IBP  | 0.771 | 0.076 | 0.092 | 0.130 | 0.743 | 0.040 | 0.090 | 0.178 | 0.699 | 0.162 | 0.253 | 0.288 |
| L-DIF  | 0.763 | 0.035 | 0.051 | 0.095 | 0.738 | 0.025 | 0.051 | 0.091 | 0.698 | 0.101 | 0.128 | 0.136 |
| U-DIF  | 0.725 | **0.002** | **0.002** | **0.042** | 0.724 | **0.000** | **0.006** | **0.067** | 0.681 | **0.000** | **0.042** | **0.071** |

Table 1: Performance of each training method across three folktables datasets. For each dataset we give the accuracy (Acc, ↑), local IF violation (LFC, ↓), empirical DIF violation (E-DFC, ↓), and the adversarial DIF violation (A-DFC, ↓). We observe a consistent drop in accuracy across datasets as we enforce more strict DIF constraints. However, we notice orders of magnitude decrease in the individual fairness violation across all three metrics as methods impose stricter constraints.

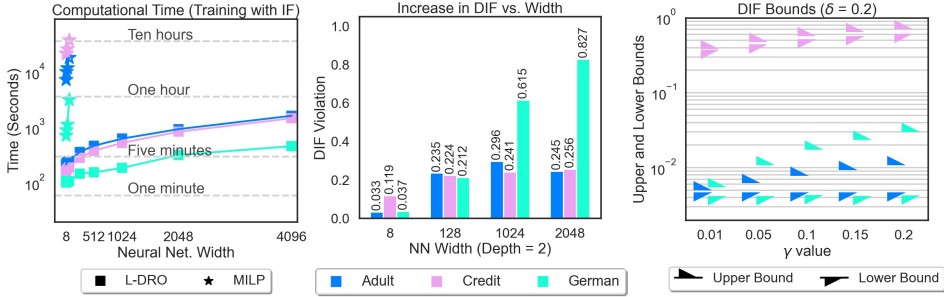

Figure 2: Empirical algorithm analysis for our upper and lower bounds on DIF. **Left:** Computational time comparison between MILP training (times reported in (Benussi et al., 2022)) and L-DIF demonstrates our methods considerable advantage in scalability. **Center:** As the NNs get larger our DIF certificates get looser, as expected with bound propagation methods (Gowal et al., 2018). **Right:** As we increase $\gamma$ we also increase the gap between our upper and lower bounds due to the over-approximate nature of our upper-bounds.

decrease in accuracy, but achieve an order of magnitude decrease in the local and distributional fairness violations. Using both L-DIF and U-DIF results in the strongest IF and DIF guarantees, albeit at a larger accuracy cost. Each regularization method proposed provides a different fairness-accuracy trade-off which should be carefully chosen to suit the needs of a given application.

## 6.2 Empirical Bound Analysis

In this section, we provide an overview of our algorithm's performance as we vary architecture size and the distributional parameter $\gamma$. In the left-most plot of Figure 2, we use stars to plot the computational times reported to train a single hidden layer neural network from (Benussi et al., 2022). We use squares to denote the amount of time used by our most computationally demanding method (L-DIF). Our method scales up to 4096 neurons without crossing the one hour threshold. We report an extended version of this experiment in Appendix B. Moreover, the results from this paper were run on a laptop while the timings from (Benussi et al., 2022) use a multi-GPU machine.

In the center plot of Figure 2, we show how the value of our upper bound on the DIF violation grows as we increase the size of the neural network. As the neural network grows in size, we observe a steady increase in our upper-bounds. This is not necessarily because the learned classifier is less individually fair. The steady increase in the bound value can be attributed to the approximation incurred by using bound propagation methods. It is well-known that as the neural networks grow larger, certification methods become more approximate (Gehr et al., 2018; Wicker et al., 2021).

In the right-most plot of Figure 2, we show how our upper and lower bounds respond to increasing the distributional parameter $\gamma$. As $\gamma$ grows, we expect that our bounds become loose due to the over-approximation inherent in our bounds. Here we consider a two-layer 16 neuron per-layer NN. Indeed we observe in the right-most plot of Figure 2, that the gap between our upper and lower bounds grow to almost an order of magnitude when we increase $\gamma$ from 0.01 to 0.2.

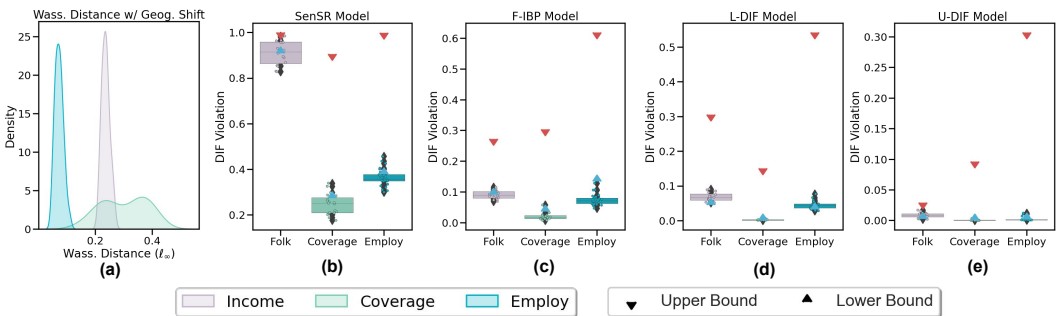

Figure 3: Evaluating our bounds versus real-world distribution shifts. **Column (a):** An empirical distribution of Wasserstein distances between the distribution of individuals from different pairs of states, we certify w.r.t the upper quartile of these distributions. **Columns (b) - (e):** We plot the local fairncess certificates (LFC) for each of the shifted dataset using a boxplot. We then plot our lower bound on the worst-case DIF violation as a blue triangle and our upper bound on the worst-case DIF violation as a red triangle.

## 6.3 Certification for Real-World Distribution Shifts

In this section we compare our adversarial bounds on DIF against real-world distribution shifts. In Figure 3 we study our ability to bound a shift in geographical context. We use the folktables dataset (Ding et al., 2021) and assume we can only train our model using data from California. We would then like to ensure that a our model is individually fair when applied to other states. We start by computing the Wasserstein distance between the data observed for different states. The distribution of these values is plotted in Figure 3 (a). By estimating the pairwise Wasserstein distance between the states data we can estimate what $\gamma$. In practice, one would use domain knowledge to estimate $\gamma$. Each dataset has a different worst-case $\gamma$ value, i.e., 0.15 for Employ versus 0.4 for Coverage. In Figure 3 (b)-(e) we then compute our upper bounds (red triangle) and lower bounds (blue triangle) for each observed value of $\gamma$. We also plot the observed local individual fairness certificates centered at each of the shifted datasets, i.e., the local fairness of our model when applied to North Carolina etc., and plot this as a box plot. We observe that our lower-bound tightly tracks the worst-case observed individual fairness. Our upper-bound is over-approximate but reasonably tight, especially for NNs trained with DIF guarantees. The validation of our upper and lower bounds against real world DIF violations highlights the value of our bounds as a scalable, practical, and sound source of guarantees for individual fairness such that model stakeholders can justify deployment.

## 7 Conclusion

In conclusion, our proposed method addresses the crucial need for scalable, formal guarantees of individual fairness. Our novel convex approximation of IF constraints enables efficient real-time audits of individual fairness by significantly reducing the computational cost local IF certification. We introduced the first certified bounds on DIF and demonstrated their applicability to significantly larger neural networks than previous works. Our empirical investigation of real-world distribution shifts further validated the scalability, practicality, and reliability of the proposed bounds. Our bounds on DIF violations offer a robust source of guarantees for individual fairness, enabling model stakeholders to justify the deployment of fair machine learning algorithms.

**Limitations** This work improves the scalability of prior IF guarantees by more than two orders of magnitude. However, the models we consider are smaller than models that may be deployed in practice we hope to further improve scalability in future work.

**Broader Impact** This work focuses on fairness quantification and bias reduction which is one critical aspect of developing socially responsible and trustworthy machine learning (ML). We highlight that this work focuses on individual fairness which is only one aspect of fairness in machine learning and on its own does not constitute a complete, thorough fairness evaluation. Experimental analysis of the impact of DIF on group fairness is in Appendix B.4. The intention of this work is to improve the state of algorithmic fairness in ML, and we do not foresee negative societal impacts.

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
