| Dataset | Depth | Width | Learning Rate | Epochs | Delta | Gamma | Optimizer |
|---------|-------|-------|---------------|--------|-------|-------|-----------|
| Adult | 2 | 256 | 0.001 | 50 | 0.02 | 0.025 | Adam (momentum=0.9) |
| Credit | 2 | 256 | 0.01 | 50 | 0.02 | 0.025 | Adam (momentum=0.9) |
| German | 2 | 256 | 0.0025 | 50 | 0.02 | 0.025 | Adam (momentum=0.9) |
| Income | 2 | 256 | 0.001 | 50 | 0.02 | 0.025 | Adam (momentum=0.9) |
| Coverage | 2 | 256 | 0.001 | 50 | 0.02 | 0.025 | Adam (momentum=0.9) |
| Employ | 2 | 256 | 0.001 | 50 | 0.02 | 0.025 | Adam (momentum=0.9) |

Table 2: Hyperparameters for the base model of each dataset. We keep all parameters across models the same with the exception of learning rate which is fine-tuned according to best accuracy on a validation set.

# A   Experimental Details

In this section, we report the hyperparameters of each base model used in our paper, details in Table 2. The only hyperparameter that is tuned is done per dataset using a 10% validation split. The best learning rate is then chosen after a grid search over 20 evenly spaced values. We have chosen a model architecture that is considerably larger than what is used by previous certification works in order to understand if the claimed scalability benefits of our method holds in practice over a half dozen datasets and compared to real-world distribution shifts. Indeed, we find in the main text that despite our base model being considerably larger than previous works, we are able to get strong IF and DIF guarantees despite the models size. In our further experiments, we keep all hyperparameters constant unless otherwise noted. For example, in our exploration of increased width we only vary the width holding all other hyperparameters in Table 2 constant.

# B   Additional Experimental Ablations

In this section, we report further experimental abalations that validate the effectiveness of our proposed method. We start by studying the empirical convergence of our approach to the true global optimal value which is computed numerically. We then provide an extended analysis of a real world distribution shift which stems from population shift over time (specifically between 2015 and 2022). Finally, we report on the scalability of our proposed F-IBP training method where we find that it has remarkable scalability benefits.

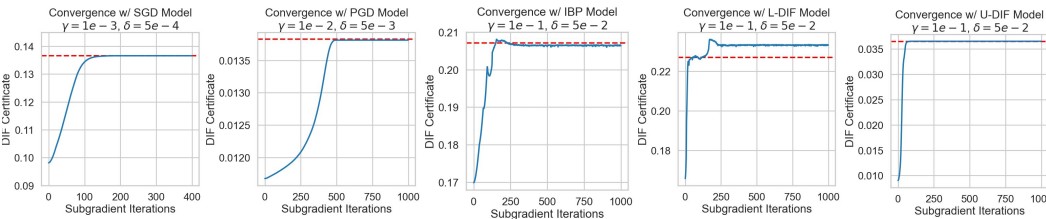

Figure 4: Convergence analysis for subgradient methods solution to Equation (5). Blue line plots the value of the optimization problem over a fixed number of iterations while the red dashed line indicates the numerically optimal solution to the optimization problem. We consider the Income dataset on networks trained with various levels of DIF regularization. From left to right we run our convergence analysis on neural networks trained with SGD, SenSR, F-IBP, L-DIF, and U-DIF regularization.

|         | Numerical Error |         |          |
|---------|-----------------|---------|----------|
|         | Income          | Employ  | Coverage |
| SGD     | -0.0143         | 0.03706 | 0.113219 |
| SenSR   | 0.06885         | 0.03659 | -0.033412 |
| F-IBP   | 0.00820         | -0.00098 | 0.022244 |
| L-DRO   | 0.05840         | -0.00132 | 0.018893 |
| U-DRO   | -0.0021         | 0.008271 | -0.007846 |

Table 3: For each combination of regularization method and Folktables dataset we compare the error between a subgradient solution to Equation (5) and the numerically optimal value. Positive entries indicates that the solution found by the subgradient method was greater than that of the numerical algorithm and visa versa for negative entries. We highlight that on average the actual value of the optimization was roughly 0.3, therefore error on the order of 1e-2 is acceptable.

## B.1 Convergence Experiments

In Section E we provide proofs that solving for a global optimum of Equation (5) is a true upper bound to the DIF violation and that any suitable subgradient algorithms should converge to a global optimum in finite time. In this Section, we discuss the experimental convergence of our U-DIF algorithm to the global optimum. In order to approximately compute the true global optimum, we use the following numerical scheme. We assume we would like to compute the DIF violation according to $\delta = 0.1$ and $\gamma = 0.05$ (exact numbers vary by network and are given in Figure 4). We further assume that we are given 50 individuals to compute the DIF i.e., $n = 50$. We start by evaluating the function $\overline{\mathcal{I}(f^\theta, x^{(i)}, \delta + \varphi^{(i,k)})}$ for each individual and for each $k \in [K]$ (we choose $k = 500$) values of $\varphi$ evenly spaced between 0.0 and 2.4 as this (more generally $(n\gamma) - \delta$) is an upper bound to the radius that any one individual can have without breaking the Wasserstein constraint. After evaluating $\overline{\mathcal{I}(f^\theta, x^{(i)}, \delta + \varphi^{i,k})}$ for each of the $k$-many $\varphi^{(i,k)}$ values for all individuals, we exhaustively enumerate all possible assignments of $\{\varphi^{(i,k)}\}_{i=1}^{50}$ such that $1/50 \sum_{i=1}^{50} \varphi^{(i)}$ is less than or equal to 0.05 ($\gamma$). As $K \to \infty$, this produces an exact computation of the global maximum of Equation (5).

Once a numerical solution has been computed, we run our U-DIF optimization using the method of Hu et al. (2020) and we plot the results for the Income dataset in Figure 4. We find that regardless of how the neural network is trained, our algorithm quickly converges to the global optimum assignment of $\{\varphi^{(i)}\}_{i=1}^n$. We note that in the center plot of Figure 4, corresponding to the F-IBP trained network, we converge to a value slightly below the global optimum; however, this is within the expected convergence error. In order to better understand this expected convergence error, we run the experiment from Figure 4 across each of the folktables datasets, and report the numerical error of our U-DIF computed value in Table 3. Only in cases where the value reported is negative do we converge to something smaller than the value produced by the numerical solution, and in each case the numerical error of the subgradient procedure converges to within tolerable error as theoretically expected.

## B.2 Extended results on real-world shifts

In this Section we briefly describe Figure 5 which presents a complementary empirical study to that of Figure 3 in the main text. In the far left hand side of Figure 5 we plot the Wasserstein distance between pairs of datasets that differ only in the year the data was collected. For the Folktables datasets this varies between 2015 and 2021. We find that the data shift over time is less severe than the data shift over geographic location. Again we use the upper quartile of these distributions as the value that we would like to certify. In Figure 5 (b)-(e) we plot the distribution of IF violation for an observed temporal shift as a box plot. For each dataset, we observe that more strict DIF regularization corresponds to better performance with respect to empirical DIF violations. Moreover, in Figure 5 (b)-(e) we plot our computed upper and lower DIF bounds. We find, as in the main text, that our lower-bound tightly tracks the empirically observed worst-case DIF violation, while our upper bound is a sound over-estimate of the DIF violation.

| | Income | | | | Employ | | | | Coverage | | | |
|---|---|---|---|---|---|---|---|---|---|---|---|---|
| | D. Par. | Eq. Od | Eq. Op | IF. Par | D. Par | Eq. Od | Eq. Op | IF. Par | D. Par | Eq. Od | Eq. Op | IF. Par |
| SGD | 0.074 | **0.027** | 0.006 | **0.000** | 0.012 | 0.103 | 0.003 | 0.011 | **0.006** | **0.023** | **0.0082** | 0.012 |
| SenSR | 0.009 | 0.139 | 0.037 | 0.021 | 0.049 | 0.156 | 0.017 | 0.041 | 0.055 | 0.112 | 0.074 | 0.009 |
| F-IBP | 0.013 | 0.081 | 0.014 | **0.000** | 0.030 | 0.119 | **0.0009** | 0.042 | 0.033 | 0.070 | 0.055 | 0.008 |
| L-DIF | 0.021 | 0.157 | 0.020 | **0.000** | 0.016 | 0.092 | 0.001 | 0.042 | 0.032 | 0.069 | 0.054 | 0.006 |
| U-DIF | **0.000** | 0.094 | **0.004** | **0.000** | **0.009** | **0.082** | 0.002 | **0.000** | 0.028 | 0.064 | 0.054 | **0.000** |

Table 4: Affect of IF training on measures of group fairness. For each metric lower is better. We specifically provide computations of demographic parity (D. Par), equalized odds (Eq. Od), equalized opportunity (Eq. Op), and IF parity (IF. Par). More often than not, we find that DIF training improves measures group fairness; though in some instances makes group fairness considerably worse.

### B.3 Extended scalability results

In Figure 6 we report the computational time required by our proposed F-IBP method as well as the computational times reported by Benussi et al. (2022). We highlight that the times reported by Benussi et al. (2022) utilize a multi-GPU machine while our experiments are conducted using only a consumer-grade laptop. Despite a considerable compute disadvantage, our methods never takes longer than 5 minutes to train neural networks with more than 4000 neurons whereas the training required by the MILP procedure can take up to 10 hours to train a 64 neuron NN. This impressive scalability improvement underscores the value of our proposed method.

### B.4 Impact on Group Fairness

While individual fairness is a flexible and key measure of fairness, it is currently the case that no one fairness metric alone captures a complete picture of model bias (Fazelpour & Lipton, 2020). In this section, we briefly report on the effect of our proposed training method on group fairness notions. In Table 4, we report common group fairness notions such as demographic parity, equalized odds, and equalized opportunity for each of our trained models. Additionally we consider IF parity which is taken as the difference in average local individual fairness violation of the model w.r.t the majority group (men) and the minority group (women). Across these metrics, we find that more often than not DIF training also has a positive effect on group fairness. While this is not always the case, Table 4 clearly establishes that DIF training methods do not necessarily exacerbate other forms of model bias.

## C Extended Related Works

In the main text, Section 2 describes the context of our contribution relative to the literature on guarantees for individual fairness (local and global) as well as works related to distributionally robust fairness. In this section, we describe some works in distributional robustness that have a strong relationship to our contribution. In Sinha et al. (2017) the authors provide an initial certificate of distributional robustness under many of the same assumptions as our method e.g., restricting to Wasserstein distances between Dirac measures. However, their bound is derived based on a Taylor expansion and relies upon a knowing the value of global Lipschitz constants of the neural network. Our method, on the other hand, makes no such assumptions on the value of Lipschitz constants. Indeed, our bound relies on the Hölder continuity of the $\mathcal{I}$ function; however, we do not rely on any knowledge of value of the constants for which the Hölder condition holds. The requirement of the Hölder condition is a practical consideration in order to ensure that our bounds converge globally with subgradient methods. Using branch and bound would solve our optimization problem without any continuity assumption on $\mathcal{I}$, albeit at greater computational complexity. For further works focusing on distributional robustness when using the Wasserstien distance, we point interested readers to Blanchet & Murthy (2016) and Mohajerin Esfahani & Kuhn (2018). Since the submission of this work, (Doherty et al., 2023) has investigated individual fairness in the context of Bayesian neural networks and find that uncertainty benefits individual fairness, perhaps due to its relationship to adversarial robustness (Carbone et al., 2020).

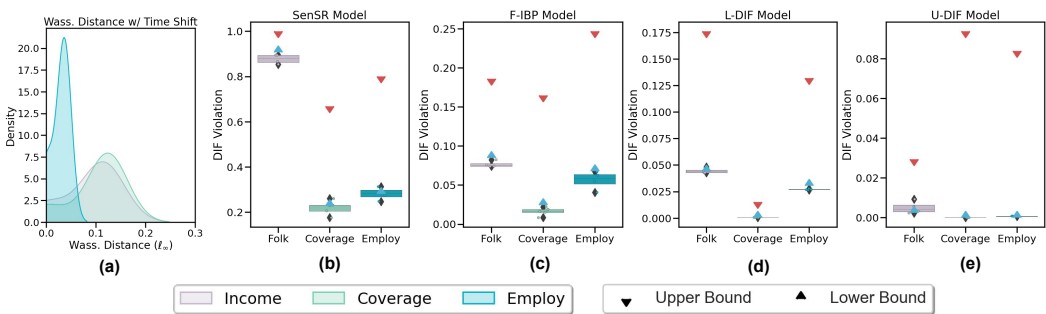

Figure 5: Evaluating our bounds versus real-world distribution shifts. **Column (a):** An empirical distribution of Wasserstein distances between the distribution of individuals from different pairs of years, we certify w.r.t the upper quartile of these distributions. **Columns (b) - (e):** We plot the local fairncess certificates (LFC) for each of the shifted dataset using a boxplot. We then plot our lower bound on the worst-case DIF violation as a blue triangle and our upper bound on the worst-case DIF violation as a red triangle.

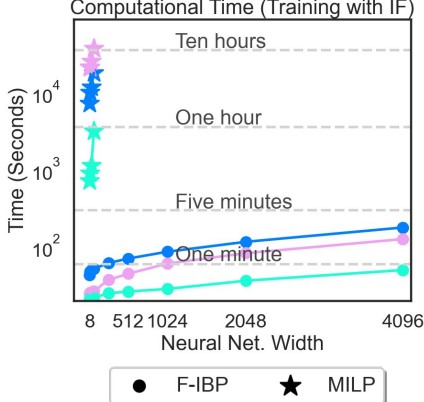

Figure 6: Computational time comparison between MILP training times reported in (Benussi et al., 2022) (plotted as stars) and F-IBP (plotted as circles) demonstrates our methods considerable advantage in scalability.

## D    Detailed Computations

In this section we first describe how fair distance metrics are computed. We then provide the details for the exact computations of IBP and finally, an analysis of the bound presented in Theorem 5.1.

### D.1    Computing Fair Distance Metrics

In this subsection we breifly describe methods to compute $d_{\text{fair}}$ metrics. Below, we describe two popular forms of $d_{\text{fair}}$ and how they are computed. In this work, we assume that the $d_{\text{fair}}$ is always given by the SenSR metric.

**Weighted $\ell_p$ metric**    A weighted $\ell_p$ metric for a vector $x$ in $\mathbb{R}^n$ takes the form: $\left( \sum_{i=0}^{n-1} \phi_i x^p \right)^{1/p}$ where $\phi$ is the weight vector. In John et al. (2020) the $\phi_i$ is set to 0 for sensitive features and 1 for non-sensitive features. In this work, we attempt to capture the intra-correlation between sensitive and non-sensitive features by setting the weight of non-sensitive features $i$ to be $\phi_i = 1/|\rho_{i,j}|$ where $\rho_{i,j}$ is the Pearson correlation coefficient between the feature $i$ and the sensitive feature $j$.

**SenSR metric**    While weighted $\ell_p$ metrics are intuitive they may be seen as overly simple. In Yurochkin et al. (2019), the authors propose the SenSR metric (abbreviated as SR). Assuming only a single sensitive attribute, the SR metric is a Mahalanobis metric computed by first learning a logistic

regression model to predict the sensitive attribute ($x_{\text{sens}}$) from the $n - 1$ non-sensitive attributes ($x_{\text{nonsens},i} \forall i \in [n-1]$), e.g., $\hat{x}_{\text{sens}} = exp(a_j^T x_{\text{nonsens}} + b_j)/\sum_{j=0}^{K} exp(a_j^T x_{\text{nonsens}} + b_j)$. Taking each vector $a_j$ to be the column of a matrix $A$, we have that the *sensitive subspace* matrix $S$ is given by $S = I - P_{ran(A)}$ where $P_{ran(A)}$ is the orthogonal projector of the span of $A$. We then take $S$ to be the matrix for a fair Mahalanobis metric $d_{\text{fair}} = d_S(x, y) = \sqrt{(x - y)^\top S^{-1}(x - y)}$.

### D.2 IBP Computations

Given a NN $f^\theta$ and an interval $[x^L, x^U]$ computed according to Theorem 5.1, interval bound propagation proceeds by computing an interval over outputs $[y^L, y^U]$ such that $\forall x \in [x^L, x^U]$, $y^L \leq f^\theta(x) \leq y^U$. Ultimately, computing the largest difference inside of $[y^L, y^U]$ will allow us to certify local IF. The output bounds, $[y^L, y^U]$, can be computed by performing a forward pass through the neural network with interval bound propagation (IBP). We adopt the notation for IBP proposed in Gowal et al. (2018) where $z^{k,L}$ and $z^{k,U}$ are the lower and upper bound on the inputs to the $k^{th}$ layer of the neural we can propagate this interval from layer $k$ to $k + 1$ as follows:

$$z_\mu^k = (z^{k,L} + z^{k,U})/2, \quad z_r^k = (z^{k,U} - z^{k,L})/2 \tag{6}$$

$$\zeta_\mu^k = W^{k+1} z_\mu^k + b^{k+1}, \quad \zeta_r^k = |W^{k+1}| z_r^k \tag{7}$$

$$z^{k+1,L} = \sigma(\zeta_\mu^k - \zeta_r^k), \quad z^{k+1,U} = \sigma(\zeta_\mu^k + \zeta_r^k) \tag{8}$$

By passing the input interval $[x^L, x^U]$ through the above equations, as $z^{0,L}$ and $z^{0,U}$ respectively, we arrive at sound upper and lower bounds of the logits of the network. For the final activation, in our case the softmax, we can compute the lower and upper bounds on the softmax output for class $i$ with:

$$\sigma_i^{K,L} = \frac{e^{z_i^{K,L}}}{e^{z_i^{K,L}} + \sum_{j \neq i} e^{z_j^{K,U}}}, \tag{9}$$

$$\sigma_i^{K,U} = \frac{e^{z_i^{K,U}}}{e^{z_i^{K,U}} + \sum_{j \neq i} e^{z_j^{K,L}}} \tag{10}$$

Finally, we have that for a $c$-class classification network that $\max_{i \in [c]} \left( \sigma_i^{K,U} - \sigma_i^{K,L} \right) \leq \epsilon$ implies that local individual fairness is satisfied and constitutes a local individual fairness certificate for the network $f^\theta$ at $x$.

### D.3 IBP Analysis

Any non-exact approximation of S can grow exponentially loose as the number of positive eigenvalues increases. For instance, even if we approximate S with a hyper-rectangle aligned with the major axes of S, as done in (Benussi et al., 2022; Ruoss et al., 2020), the ratio of volume of the $\delta$ Mahalanobis ball ($\{x \in \mathbb{R}^n \mid d_S(x) \leq \delta\}$) and volume of hyper-rectangle is given by $\frac{\pi^{d/2}}{\Gamma(d/2+1)} \prod_{j|\lambda_j > 0} \lambda_j$ to $4^{d/2} \prod_{j|\lambda_j > 0} \lambda_j$, where $\lambda_j$ is the $j^{th}$ eigenvalue of $S$, and $d$ is the number of positive eigenvalues of S. It is not hard to see that the ratio of volumes decrease super-exponentially. Our method, an approximation aligned with the cannonical axis shares this quality. We confirm the super-exponential decrease in volume ratio with Monte-Carlo estimation in Figure 7. Since this ratio decreases super-exponentially, the constraint imposed by an approximating orthotope could be too stringent when $d$ is large.

## E   Proofs

In this section, we first prove Theorem 5.1. We then show how Equation (1) and Equation (3) are equivalent given that the Wasserstein ball is over Dirac measures. Next, we provide a straight-forward proof that a global optima for Equation (5) is a sound upper bound on Equation (3). Lastly, we prove that the local IF function, $\mathcal{I}$ is Hölder continuous, therefore suitably chosen subgradient methods should converge to the global solution.

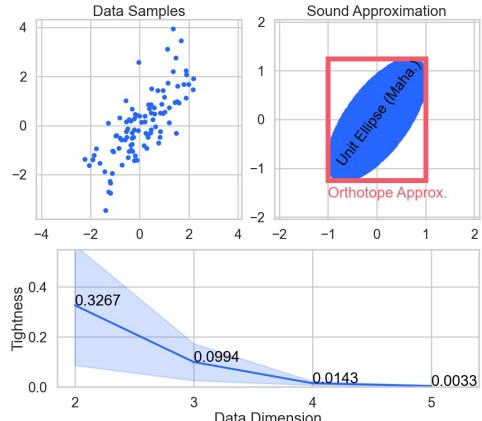

Figure 7: Visualization of our orthotope approximation. **Top Left:** Synthetic 2D data samples. **Top Right:** Unit ellipse with respect to the Mahalanobis distance computed on the data (blue) and our over-approximation of this ellipse (red). **Bottom:** The tightness (ratio of unit ellipse volume and approximation volume) computed for 20 random matrices with different dimensions. The super-exponential decrease of this approximation is discussed in Section D.3.

## E.1 Proof of Theorem 5.1

**Theorem E.1.** *Given a positive semi-definite matrix $S \in \mathbb{R}^{m \times m}$, a feature vector $x' \in \mathbb{R}^m$, and a similarity theshold $\delta$, all vectors $x'' \in \mathbb{R}^n$ satisfying $d_S(x', x'') \leq \delta$ are contained within the axis-aligned orthope:*

$$\left[ x' - \delta\sqrt{d}, x' + \delta\sqrt{d} \right]$$

*where $d = diag(S)$, the vector containing the elements along the diagonal of $S$.*

*Proof.* Consider the case of $\delta = 1$. The width of the desired interval along the $i^{th}$ dimension can be obtained by solving $\max_{x''} e_i^T(x'' - x')$ *s.t.* $d_S(x', x'') \leq 1$ where $e_i$ is the $i^{th}$ canonical basis vector. Let $S = R^T R$ where $R$ is the matrix square root of $S$, we can re-write the optimization in the rotated space by change of variables, $u := R^T(x'' - x')$. W.r.t. $u$ the optimization becomes $\max_u R_i^T u$ *s.t.* $||u|| \leq 1$ where $R_i$ is the $i^{th}$ column of $R$. The solution of this optimization problem is then $R_i^T R_i / ||R_i|| = \sqrt{R_i^T R_i} = \sqrt{S_{i,i}}$. Generalizing to each dimension $i$ we have the bound $\sqrt{diag(S)}$, as desired. Because $S^{-1}$ is a linear transformation this bound remains sound when scaled by $\delta$ or translated to be centered at an arbitrary feature vector $x$. Proof of results related to Theorem 5.1 can be found in Emrich et al. (2013); Alger (2020). $\square$

**Lemma 2.** *Computing IBP w.r.t. the interval $[x' - \delta\sqrt{d}, x' + \delta\sqrt{d}]$ from Theorem 5.1 results in a sound over-approximation of the individual fairness violation as defined in Definition 1.*

*Proof.* IBP produces output bounds $[y^L, y^U]$ such that $\forall x'' \in [x' - \delta\sqrt{d}, x' + \delta\sqrt{d}], f^\theta(x'') \in [y^L, y^U]$. By taking $\bar{\epsilon} = |y^U - y^L|$, we can certify that $\forall x''$ *s.t.* $d_{\text{fair}}(x', x'') \leq \delta \implies |f^\theta(x') - f^\theta(x'')| \leq \bar{\epsilon}$ which is exactly the condition needed to prove that local individual fairness holds according to Definition 1. $\square$

## E.2 Equivalence between Equation (1) and Equation (3) for Dirac Measures

Here we briefly describe the equivolence between Equation (1) and Equation (3) when both distributions are Dirac measures, i.e., distributions given by a set of observed samples $\{x^{(i)}\}_{i=1}^n$. In this case, the reference distribution, $\hat{P}$, has support $\{\boldsymbol{\delta}_{x^{(i)}}\}_{i=1}^n$ where $\boldsymbol{\delta}_{x^{(i)}}$ is the Dirac function at location $x^{(i)}$, and we assume each observed sample has equivalent probability $1/n$. We then would like to measure the $p-$Wasserstein distance between $\hat{P}^0$ and another distribution. We make the assumption

that the other distribution is also a Dirac measure. In this case, the $p-$Wasserstein distance between $\hat{P}$ and another Dirac measure $\hat{Q}$ with samples $\{z^{(i)}\}_{i=1}^{n}$ is known to be given by:

$$W_p(\hat{P}, \hat{Q}) := \inf_{\pi} \Big( \sum_{i=1}^{n} ||x^{(i)} - z^{\pi(i)}||^p \Big)^{1/p}$$

where $\pi$ is the set of all permutations of the set $[n]$. Without loss of generality, one can express $\hat{Q}$ as being given by a set of perturbation vectors $\phi^{(i)}$ such that the elements of $\hat{Q}$ are defined to be $\{x^{(i)} + \phi^{(i)}\}_{i=1}^{n}$. In this case, it is clear that the $p-$Wasserstein distance is simply $\sum_{i=1}^{n}(||\phi^{(i)}||^p)^{1/p}$. Further, this makes it straight-forward to see that all distributions $\hat{Q}$ that are within a $p-$Wasserstein ball around $\hat{P}$ satisfy the constraint that $\sum_{i=1}^{n}(||\phi^{(i)}||^p)^{1/p} < \gamma$. Taking both sides to the $p^{th}$ power, we notice that this is precisely the constraint in Equation (1). Thus, the $\hat{Q}$ maximizing Equation (3) corresponds to the Dirac measure maximizing Equation (1), as desired.

**Alternate and general proof**

**Theorem E.2.** *Inputs from a distribution that is within $\gamma$ p-Wasserstein distance from the source empirical distribution $\hat{P}(\mathbf{x}) = \sum_{i=1}^{n} \delta(\mathbf{x} = \mathbf{x}_i)$ must be contained in the region given by $\sum_{i=1}^{n} B(\mathbf{x}_i, \delta_i)$ where $\delta_i > 0, \sum_{i=1}^{n} \delta_i^p \leq \gamma^p$. In other words, the constraint in (3) over-constraints and therefore satisfies the constraints of (1).*

*Proof.* Target distributions $Q$ that are $\gamma$ distance within the source distribution must satisfy the following constraint for all possible joint distributions: $\tau(\hat{P}, Q)$ over distributions $\hat{P}, Q$.

$$W_p(\hat{P}, Q) = \left( \inf_{\kappa \sim \tau(\hat{P}, Q)} \mathbb{E}_{(x,y) \sim \kappa}[d(x, y)^p] \right)^{1/p}$$

Since $\hat{P}$ is an empirical distribution, minimum over joint distributions is obtained when an instance $y$ sampled from $Q$ is coupled with a closest point in $\hat{P}$ denoted $q(y, \hat{P}) = \arg\min_{i \in [1,n]} d(\mathbf{x}_i, y)$. We therefore have the following simplified expression for $W_p$.

$$W_p(\hat{P}, Q) = \mathbb{E}_{y \sim Q}[d(q(y, \hat{P}), y)^p]^{1/p}$$

We denote by $\delta_i$ the maximum distance between $\mathbf{x}_i$ and any point that is sampled from $\mathbf{Q}$ that is coupled with $\mathbf{x}_i$, i.e.

$$\delta_i \triangleq \max_{y \sim Q} d(q(y, \hat{P}) = \mathbf{x}_i, y)$$

The definition of $\delta_i$ leads to an upper bound on the $W_p$ distance as follows.

$$\begin{aligned}
W_p(\hat{P}, Q) &= \mathbb{E}_{y \sim Q}[d(q(y, \hat{P}), y)^p]^{1/p} \\
&\leq \sum_i \left( \Pr(q(y, \hat{P}) = \mathbf{x}_i) \delta_i^p \right)^{1/p} \\
&\leq \left( (\sum_i \Pr(q(y, \hat{P}) = \mathbf{x}_i))(\sum_i \delta_i^p) \right)^{1/p} \\
&= (\sum_i \delta_i^p)^{1/p}
\end{aligned}$$

Following the definition of $\delta_i$, the support of $Q$ (i.e. set of all points with non-zero probability) must be contained in $\cup_{i=1}^{n} B(\mathbf{x}_i, \delta_i)$, which in itself is contained in $\sum_i B(\mathbf{x}_i, \delta_i)$ with the additional constraint that $\sum_i \delta_i^p \leq \gamma^p$ □

### E.3 Proof that Global Maximum of Equation (5) is a Certificate

Recall:

**Theorem 5.2** Given an optimal assignment of $\{\varphi^{(i)}\}_{i=1}^n$ in Equation (5), the corresponding $\bar{\epsilon}$ is a sound upper-bound on the DIF violation of the model and therefore, is a certificate that no $\gamma$-Wasserstien distribution shift can cause the individual fairness of the model to exceed $\bar{\epsilon}$.

*Proof.* Let the set of vectors $\{\phi^{\star(i)}\}_{i=1}^n$ represent the global maximizing assignment of Equation (3). Let $\{\varphi^{\star(i)}\}_{i=1}^n$ be the set of real values such that $\varphi^{\star(i)} = ||\phi^{\star(i)}||$. From Theorem 5.2 we have that for each $i$ that $\mathcal{I}(f^\theta, x^{(i)} + \phi^{\star(i)}, \delta) \leq \overline{\mathcal{I}(f^\theta, x^{(i)}, \delta + \varphi^{\star(i)})}$, thus we have that $\sum_{i=1}^n \mathcal{I}(f^\theta, x^{(i)} + \phi^{\star(i)}, \delta) \leq \sum_{i=1}^n \overline{\mathcal{I}(f^\theta, x, \delta + \varphi^{\star(i)})}$. Given that $\{\varphi^{\star(i)}\}_{i=1}^n$ is a feasible assignment of the optimization problem and that it upper-bounds the maximum of Equation (3), we have that the global maximum of Equation (5), must be an upper bound on Equation (3). This is due to the fact that either $\{\varphi^{\star(i)}\}_{i=1}^n$ is the maximizing assignment, or a the global maximum returns a value larger than that returned by $\{\varphi^{\star(i)}\}_{i=1}^n$ which would also be an upper bound to Equation (3). $\square$

### E.4 Proof $\mathcal{I}$ is Hölder Continuous (therefore globabally converges)

Let $\mathbf{x}_i$ denote a training instance in the original distribution, $\mathbf{x}_i^{(1)}, \mathbf{x}_i^{(2)}$ denote the points in the neighbourhood of $\mathbf{x}_i$ such that

$$\mathbf{x}_i^{(1)} = \mathbf{x}_i + \varphi_i$$
$$\mathbf{x}_i^{(2)} = \arg\max_{\mathbf{x} \in B_\delta(\mathbf{x}_i^{(1)})} |f(\mathbf{x}) - f(\mathbf{x}_i^{(1)})|.$$

We have the following inequality for the absolute difference between function values at $\mathbf{x}_i^{(1)}$ and $\mathbf{x}_i^{(2)}$.

$$|f^\theta(\mathbf{x}_i^{(1)}) - f^\theta(\mathbf{x}_i^{(2)})| \leq |f^\theta(\mathbf{x}_i^{(1)}) - f^\theta(\mathbf{x}_i) + f^\theta(\mathbf{x}_i) - f^\theta(\mathbf{x}_i^{(2)})|$$
$$\leq |f^\theta(\mathbf{x}_i^{(1)}) - f^\theta(\mathbf{x}_i)| + |f^\theta(\mathbf{x}_i) - f^\theta(\mathbf{x}_i^{(2)})|$$
$$\leq 2I[f^\theta; \mathbf{x}_i, \varphi_i + \delta]$$

which follows because $|\mathbf{x}_i^{(2)} - \mathbf{x}_i| \leq |\mathbf{x}_i^{(1)} - \mathbf{x}_i| + |\mathbf{x}_i^{(2)} - \mathbf{x}_i| \leq \varphi_i + \delta$

where I is as defined above

$$\text{i.e.} I[f^\theta; \mathbf{x}, \delta] \triangleq \max_{\hat{\mathbf{x}} \in B_\delta(\mathbf{x})} |f^\theta(\hat{\mathbf{x}}) - f^\theta(\mathbf{x})|$$

Note that the inequality $|f^\theta(\mathbf{x}_i^{(1)}) - f^\theta(\mathbf{x}_i^{(2)})| \leq 2I[f^\theta; \mathbf{x}_i, \varphi_i + \delta]$ holds for any arbitrary value of $\mathbf{x}_i$ and $\varphi_i$.

Therefore eq3a becomes,

$$I(f^\theta; \mathbf{x}_i^{(1)}, \delta) \leq 2I[f^\theta; \mathbf{x}_i, \varphi_i + \delta]$$
$$\implies \frac{1}{n} \sum_{i=1}^n I(f^\theta; \mathbf{x}_i^{(1)}, \delta) \leq \frac{2}{n} \sum_{i=1}^n I(f^\theta; \mathbf{x}_i, \varphi_i + \delta)$$

The upper bound on the RHS can be obtained easily since $g_i(\varphi_i) = I(f^\theta; \mathbf{x}_i, \delta + \varphi_i)$ is quasiconvex, which is easily seen by noting that any monotonic function is quasiconvex.

Moreover, if f is C-lipschitz, then $\sum_i g_i(\varphi_i)$ satisfies Holder condition of order p for p=1. The proof is as follows.

$$|\sum_i g(\varphi_i^{(1)}) - \sum_i g(\varphi_i^{(2)})| \leq \sum_i |I(f^\theta; \mathbf{x}_i, \varphi_i^{(1)}) - I(f^\theta; \mathbf{x}_i, \varphi_i^{(2)})|$$

$$= \sum_i |f^\theta(\mathbf{x}_i^{(1)}) - f^\theta(\mathbf{x}_i)| - I(f^\theta; \mathbf{x}_i, \varphi_i^{(2)})$$

where WLOG, we assume $\varphi_i^{(2)} \leq \varphi_i^{(1)}$

$$\leq \sum_i |f^\theta(\widehat{\mathbf{x}_i^{(1)}}) \pm C(\varphi_i^{(1)} - \varphi_i^{(2)}) - f^\theta(\mathbf{x}_i)| - I(f^\theta; \mathbf{x}_i, \varphi_i^{(2)})$$

where $\widehat{\mathbf{x}_i^{(1)}} = \mathbb{P}_{\varphi_i^{(2)}}[\varphi_i^{(1)}]$

$$\leq \sum_i |f^\theta(\widehat{\mathbf{x}_i^{(1)}}) - f^\theta(\mathbf{x}_i)| - I(f^\theta; \mathbf{x}_i, \varphi_i^{(2)}) + C(\varphi_i^{(1)} - \varphi_i^{(2)})$$

$$\leq \sum_i I(f^\theta; \mathbf{x}_i, \varphi_i^{(2)}) - I(f^\theta; \mathbf{x}_i, \varphi_i^{(2)}) + C(\varphi_i^{(1)} - \varphi_i^{(2)})$$

$$= \sum_i C(\varphi_i^{(1)} - \varphi_i^{(2)})$$

$$= C|\varphi^{(1)} - \varphi^{(2)}|$$

where $\mathbf{x}_i^{(1)} = \underset{\mathbf{x} \in B(\mathbf{x}_i, \varphi_i^{(1)})}{\arg\max} |f^\theta(\mathbf{x}) - f^\theta(\mathbf{x}_i)|$

and $\mathbb{P}_\varphi(\mathbf{x})$ is an operator to project a vector $\mathbf{x}$ into $B(\mathbf{x}, \varphi)$

Bounded optimization of quasiconvex functions that satisfy Holder condition of order p is shown to converge to global optimum when using exact or in-exact subgradient optimization methods in Hu et al. (2020).