# OpenReview forum: "Certification of Distributional Individual Fairness"
_NeurIPS.cc/2023/Conference — NeurIPS 2023 poster_

### Official Review · Reviewer_xyNq · 2023-07-01

**Soundness:** 3 good
**Presentation:** 3 good
**Contribution:** 2 fair
**Rating:** 6
**Confidence:** 3

**Summary:**

This paper considers the problem of certifying individual fairness (IF), which is of great importance to reliable machine learning algorithms. To this end, the authors propose a novel convex relation of IF constraints that greatly reduces the computational cost. In addition, the authors propose to certify distributional individual fairness, ensuring that the neural network has guaranteed individually fair predictions for a given empirical distribution and all distributions within a $\gamma$-Wasserstein ball.

**Strengths:**

1. This paper is technically sound.
2. The extensive experiments validate the effectiveness of the proposed methods.

**Weaknesses:**

The paper studies individual fairness and distributional fairness. To my opinion, the two topics seem to be independent. However, it is possible that I misunderstand this paper. It would be better if the authors can present more relations between these topics.

**Questions:**

## Miscellaneous
1.	Line 106: feed forward $\to$ feedforward
2.	Line 168: $d$ is indeed a vector; however, the denotation $\sqrt{d}$ should be defined more specifically.

---

> ### Author Rebuttal · Authors · 2023-08-09
>
> We thank the reviewer for their comments including their clarifying points about our notation and praise regarding our experimental section. We will ensure that the notational points are adjusted in the final version of the paper
>
> ### On the separation between distributional robustness and fairness
> In general, distributionally robust fairness and individual fairness are separate concepts. This point is made on line 97 (last paragraph of the related works). However, in this work we study the intersection of these two: distributional individual fairness. We will attempt to further emphasize this in the Introduction.

---

> > ### Comment · Reviewer_xyNq · 2023-08-10
> >
> > Thank you for your response. It addresses my concern. Since I am not expert in this field, I will keep my score.

---

### Official Review · Reviewer_Lpfq · 2023-07-06

**Soundness:** 3 good
**Presentation:** 3 good
**Contribution:** 3 good
**Rating:** 6
**Confidence:** 4

**Summary:**

This paper studies formal guarantees for notions of individual fairness (IF) for predictors given by neural network models. After relaxing common definitions for IF metrics by means of $\ell_\infty$ balls (or orthotopes), they adapt methodology based on adversarial robustness to provide upper and lower bounds to the IF achieved by models on an empirical sample - and those within a $\gamma-$Wasserstein ball about it.

**Strengths:**

- This paper studies an important problem of individual fairness
- The first half of the paper, Section 3 and 4, which cover Background, the DIF definition, and problem explanation are very clear and easy to understand.

**Weaknesses:**

- The key observation and novelty in the approach is not clearly noted (See below)
- Several of the nice advantages of their method (e.g efficiency) are not explained (see below).

**Questions:**

1. Numerous times in the paper the authors say their bounds are ”efficient” because they leverage efficient methods (e.g. those based on bound propagation). While that may be true, it would be nice for the readers if they provided a brief explanation as to why these methods are efficient instead of placing everything in the appendix.
2. It seems to me that the central novelty of this paper is to upper bound a mahalanobis metric (for $d_{fair}$) with an orthotope, which is quite simple. The remaining of the paper seems to me a direct application of results and methods in adversarial robustness. While I do appreciate the observation of being able to use those tools in the context of fairness - which also constitutes novelty - I would appreciate if the authors could be very clear about what are the main technical contributions of this work.
3. Personally, I am not sure providing a section on the impact of these methods on group fairness is necessary. I’d much rather prefer a discussion on the efficiency of the bounds.
4. Figure 1 is quite confusing. What makes the blue-star individuals likely? As presented, those blue-star points do not look likely. If I understand the figure correctly, the authors should present a more balanced empirical sample together with a larger sample representing the (unobserved) population.
5. I also have problems with the fact that the authors state their goals and present their definitions in terms of expectation (e.g. as in Def 2), but simply restrict themselves to studying empirical samples. I think the presentation is misleading, because nowhere the authors really provide guarantees for the definition in Def 2 (that is, risk bounds). This is also an important limitation where the study the Wasserstein distance between distributions, as they simply regard their distribution as a one supported on Dirac functions (on the observed samples).
6. Immediately after Eq (4), the authors write that “we can optimize this bound to be tight”. I don’t think this is correct: while they can indeed optimize the bound, there’s no guarantee that the bound will be tight, as the original problem is non-concave.
7. In Section 5.4 and after presenting $\mathcal L_{F-DIF}$, the authors mention when $\gamma=0$, one recovers a local constraint on individual fairness on $x\in X$. I don’t think this is completely accurate, because again, Def. 2 is defined in expectation of $x\sim p(x)$, not simply over the empirical sample.

**Limitations:**

The authors mention that they do not foresee negative societal impacts. Maximizing upper and lower bounds is great but in doing so we don’t really know what is happening to the true fairness violation. It may be that the true fairness violation is in fact increasing which is propagating unfairness. While I understand that solving for this value is not feasible and thus appreciate the results presented, I would also like the paper to acknowledge that there are potential negative effects.

---

> ### Author Rebuttal · Authors · 2023-08-09
>
> We thank the reviewer for their valuable comments on the presentation of our paper and appreciate the detail of their review. Below, we address each point raised by the reviewer. We clear up any minor misconceptions and provide a clear action to improve the presentation of the final version of the paper. We note that we have addressed the authors point on key contributions and observations in the global response. We are more than happy to respond on any further concerns.
>
> ### On point 1:
> On line 180-191 we state that our approach to local IF certification has computational complexity of two forward passes through the network, where one forward pass maintains the upper bound, the other maintains the lower bound. On the other hand, MILP approaches take exponentially many forward passes to converge. We will elaborate on these in the revised paper. We will also add a further comment on this in the main text that also points to Figure 2 as empirical validation of this statement.
>
> ### On point 2:
> We agree with the reviewer that one central novelty of the work is to upper bound the Mahalanobis metric with an orthotope. However, we would like to underscore that the application of this result is not to any traditional notion adversarial robustness, but to distributional individual fairness. The distributional component of fairness has not been certified in any prior work. Moreover, distributional certificates for any notion (fairness or robustness) have not been provided without assuming explicit knowledge of a Lipschitz constant. To this end, our optimization approach for this problem is novel and Theorem E.2 (to be moved to the main text as Theorem 5.3) showing that the upper bound formulation can be efficiently solved is another key non-trivial result of this paper which may find impactful application outside the current work. We appreciate that this novelty could have been further underscored and in the final version we will ensure that these points are emphasized.
>
> ### On point 3:
> We agree with the reviewers that Table 2 and Section 6.4 may be moved to the Appendix to make room for further analysis. Specific details are provided in our global response.
>
> ### On point 4:
> In Figure 1, the blue points have feature values that are generally within the observed range of individuals, albeit slightly outside what we have already seen. On the other hand, the purple stars fall wholly outside the range of feature values that were seen during training. The exact location of the blue stars is taken for illustration purposes. The blue points are not meant to illustrate the unobserved population (which implies unobserved points from the same distribution) but represent points that may be drawn from a population with a slightly different distribution. The purple individuals on the other hand illustrate individuals drawn from a drastically different distribution with different support.
>
> ### On point 5:
> On line 196 of the paper, we state that the error introduced in the finite sample regime can be bounded by means of concentration inequalities (i.e., risk bounds as the reviewer alludes to). In the final version of the work, we will elaborate on this point and provide a clear lemma statement in the main text in order to fully clarify the guarantees we give. The lemma statement will be:
>
> Lemma 5.3 Given an upper-bound $\bar{\epsilon}$ computed according to Equation (5) the error introduced by using a finite sample can be bounded from above by $\tau$. That is, the estimate $\bar{\epsilon}$ is within $\tau$ of the true expectation of $\bar{\epsilon}$ with probability $1- \lambda$, as long as $n$ is at least $\dfrac{-1}{2\tau^2} log(\lambda/2) $.
>
> The proof of this Lemma is a straight-forward application of Chernoff or Hoeffding's inequality. This lemma allows us to quantify the error induced by using finite samples to approximate the expectation in Definition 2, and works for both our upper and lower bounds.
>
> ### On point 6:
> We thank the reviewer for this note, and they are correct with this point. The phrasing of this should state "we can optimize any feasible selection of $\phi_{i}$ values to be a tighter lower-bound by observing ... the function is differentiable." We stress that any feasible selection of $\phi_{i}$ yields a valid lower bound to the DIF violation and that by observing our upper bounds we can quantify how "tight" the lower bound is w.r.t. the true value.
>
> ### On Local IF constraint when $\gamma = 0$
> We believe this is a minor misconception on behalf of the reviewer. We highlight the specific phrasing used: "we recover the local IF constraint on each $x^{(i)} \in X$." This claim is that when $\gamma = 0$, $x^{(i)}$ will not be perturbed and therefore the upper-bound fairness violation is taken at $x^{(i)}$ which is identical to the local IF constraint (note this is Definition 1 not Definition 2).
>
> ### On the negative societal impacts comment
> Given that we compute upper and lower bounds on the DIF violation, we are guaranteed that the true DIF value falls between these bounds. Therefore, we do know what is happening to the true fairness violation (w.r.t DIF) up to the tightness of our bounds. See Figures 2 and 3 of the main text for illustration of our upper and lower bounds.
>
> In addition, where "true fairness violation" means bias not captured by the notion of DIF, we specifically address this in Section 6.4 where we state "it is currently the case that no one fairness metric alone captures a complete picture of model bias" and further make the point that DIF has no correlation with group fairness notions (see Table 2), thus a complete analysis of model bias should take these definitions into account. In the final version of the draft we will include this sentiment in our Broader Impact statement.

---

> > ### Comment · Reviewer_Lpfq · 2023-08-13
> > **Thank you for your comments**
> >
> > I thank the authors for their responses, which have partially alleviated my concerns.
> >
> > 1. Understood, thanks. However note that, as posed in this paper, the local certification amounts simply to a local Lipschitz analysis. Several approaches exist for this that do not require an exponential number of iterations (as the authors suggest). See e.g. [Muthukumar et al, "Adversarial robustness of sparse local lipschitz predictors"].
> >
> > 2. Understood. I think stressing on this novelty will remark the novelty of the paper further.
> >
> > 3. Thanks.
> >
> > 4. Thank you for the clarification. However, I really do not think that this figure is helpful as it stands: note that the figure and caption refers everywhere to "likely"; strictly speaking though, the blue stars are as unlikely as the purple ones, since neither of those fall in the (apparent) support of the data distribution. I think the authors might be confounding distributions with samples, and what they mean is that purple stars are samples from a distribution that is further from the distribution used during training. Again, this has nothing to do with samples being "likely".
> >
> > 5. Yes, of course, one can employ concentration and provide a finite sample approximation result, but this requires a new fresh draw of samples. This is rather trivial, or "data waste-full", and should the need for further samples should be clearly stated in their comment on this point.
> >
> > 6. Thanks.
> >
> > 7. (on the reduction when $\gamma=0$: Thanks for the answer, but I don't think I agree: if $\gamma=0$, one requires $\mathbb E_x [ \mathcal I (f,x,\delta) ]<\epsilon$. However, this is different than requiring that $\mathcal I(f,x,\delta) < \epsilon ~\forall x \in X$, as stated -- the latter is stronger. Indeed, Definition 1 is *distributional*.
> >
> > 8. On societal implications: I appreciate the answer, but again I don't agree. I understand that you can control the upper bound to the fairness violation (the worst case violation), as indeed noted in the figures. However, reducing the upper bound does not imply that the true fairness violation (say, its mean) will be reduced as a result. I understand that the upper bound might be tight, but this does not mean that it will be tight for the specific data distribution one encounters in the real world. Maybe the authors can enlighten me as to why my understanding might be incorrect.
> >
> > 9. Oh, one last comment: As I was re-reading the paper to better understand your comments, I see that immediately after the definition of ID, the authors write "...$\mathcal I (f, x,\delta  )$ to denote the function that returns the largest value of $\epsilon$ such that the local IF property holds". I think this should read *the smallest*, since the largest $\epsilon\to\infty$ is always valid but uninformative.

---

> > > ### Author Response · Authors · 2023-08-14
> > > **Thank you for the further clarifications**
> > >
> > > We are glad that our answers alleviated some of the reviewer’s concerns, and are very grateful for their response further clarifying their points and detailed reading of our paper. Below, we further clarify our responses with respect to reviewers concerns about the paper's presentation.  We are happy to answer any further questions or concerns the reviewer may have.
> > >
> > > _On computational complexity (Point 1)_
> > >
> > > We agree there are many methods for certifying Lipschitz continuity of neural networks; however, they are not suitable for local individual fairness. This is one strength of Theorem 5.1 as a contribution: it allows one to use prior robustness certification methods for local IF. On line 174 of our paper we will make it more clear that further methods beyond IBP, including the one noted by the reviewer, are suitable and we will make this change. As for the computational complexity, prior methods for certified IF training do not make use of our contribution (Thm 5.1) and therefore rely on MILP formulations to train with local IF constraints. It is compared with these methods that Thm 5.1 + IBP has a significant computational complexity advantage.
> > >
> > > _Clarifying our example figure (Point 4)_
> > >
> > > We thank the reviewer for clarifying their concern and agree this can be made more clear. In our example Figure 1, the toy dataset is drawn from a closed-form distribution [1]. Therefore, when we say likely, we can compute the true probability density under the data-generating distribution. Unfortunately, in the original version of Figure 1, we did not compute these likelihood values. In the final version of the paper, we will use a contour plot to visualize the underlying distribution and will add computed likelihoods and Wasserstein distance to further clarify the point. Though we cannot provide the contour plot here, we state the following values that will appear in the final version of the paper:
> > >
> > > Blue points probability density (sum):  0.1246, Blue points Wass. Dist ($\ell_2$): 0.195
> > >
> > > Purple points probability density (sum): 1.363e-20, Purple points Wass. Dist ($\ell_2$): 1.297
> > >
> > > _Clarifying the use of concentration inequalities (Point 5)_
> > >
> > > In any audit of a models performance (e.g., accuracy, adversarial robustness) including DIF, one uses a held-out test set. For a meaningful evaluation, the test set ought to be representative of the data distribution. Our DIF evaluation can use the same held out test-set points that is used to evaluate test-set accuracy, which is why we do not perceive the requirement to be “data waste-full”. To demonstrate the exact data requirement, observe that if we want to guarantee our estimate has error at most 0.05 with probability 0.95 then we need less than 750 test-set samples, considerably less than what exists in many test-sets.
> > >
> > > _On the $\gamma = 0$ case (Point 7)_
> > >
> > > We thank the reviewer for the clarification and agree with their point. The initial phrasing intends to convey that for a singleton set, i.e., for each $x$ in isolation, and with $\gamma = 0$ both Definition 1 and Definition 2 are equivalent; however the phrasing is ambiguous and could be read as "when $\gamma = 0$, Definition 1 and Definition 2 are equivalent constraints" the reviewer correctly points out that this is not true as Definition 1 is a deterministic constraint over each input independently and Definition 2 is a constraint on the expectation. We will remove this sentence from the final version keeping only the important aspect that the F-IBP loss corresponds to the local IF loss function used by prior works.
> > >
> > > _On societal impact (Point 8)_
> > >
> > > To clarify this point further, imagine the worst-case DIF violation of a vanilla classifier is 0.9 and the average case DIF violation is 0.3. We interpret the reviewer's concern to be that by reducing the worst-case violation using our method to 0.5 we may also bring up the average case violation to 0.4, thus making things more unfair on average. However, we assume a threshold ($\delta$) is set _a priori_ such that any DIF violation below this threshold is tolerable, e.g., the threshold is $\delta = 0.01$. Then, our certificate can formally prove the worst-case is less than 0.01 and therefore the average-case contains negligible DIF bias. However, we appreciate the concern and will include a statement at the end of Section 5.4 about the loss functions focusing on the worst-case and not the average-case. We also highlight that at deployment time, practitioners must check that the distribution they encounter is within the $\gamma$ value of our guarantee for the bound to remain sound.
> > >
> > > _On the type-o (Point 9)_ Yes, that is correct, we will edit it out. Thanks for the great catch.
> > >
> > > [1] - Line 795 of the following open-source package specifies the distribution these are drawn from: https://github.com/scikit-learn/scikit-learn/blob/7f9bad99d/sklearn/datasets/_samples_generator.py#L786

---

> > > > ### Comment · Reviewer_Lpfq · 2023-08-14
> > > > **Thanks you for your comments**
> > > >
> > > > I appreciate the authors' dedication in responding - I'm glad to see we're converging :)
> > > >
> > > > 1. I agree that the computational aspect of the contribution here is with respect to Thm 5.1 + IBP, and not strictly w.r.t. robustness certifications methods. I think the comment that the authors plan to add, clarifying the connections to other robustness certification methods, will be beneficial for the readers.
> > > >
> > > > 4. Agreed, I think modifying this figures along these lines will be useful to the reader. In particular, it's important to note that under the original distribution of both blue and purple stars is zero (or virtually zero), but their distance in terms of Wasserstein distance is different.
> > > >
> > > > 5. I agree with your comment - my point referred to the fact that data should be different from that used to e.g. solve the proposed optimization problems. I understand this might be somewhat obvious :) i simply meant that "extending these results to hold over the population requires a validation set".
> > > >
> > > > 7. Cool!
> > > >
> > > > 8. Yes, exactly. I understand that a tolerable maximum violation can/should be set a priori for a specific application, and that the resulting method can guarantee that it remains below that level. However, as the authors correctly mention, it is possible that correcting for the worst-case fairness leads to an increase in the (true) average fairness violation. I think a comment along these lines will complement the authors discussions nicely.
> > > >
> > > > 9. Cool!
> > > >
> > > > With all these, I'm happy to increase my score.

---

### Official Review · Reviewer_FRnw · 2023-07-07

**Soundness:** 3 good
**Presentation:** 2 fair
**Contribution:** 3 good
**Rating:** 7
**Confidence:** 3

**Summary:**

This paper studies the problem of individual fairness in supervised learning. The focus is on studying how to certify distributional individual fairness (IF) (individual fairness over a set of distributions close to the observed empirical data distribution) in neural networks. Prior work has focused largely on certifying global IF, which is more expensive and thus can only be applied to smaller neural networks than the proposed certification/debiasing technique. The contributions of the paper are in showing how to certify distributional IF in neural networks and then using these bounds in the training process as regularizers to debias NNs.

The main methodology for certifying IF is presented in Section 5. The first step is to certify local IF by over-approximating the similarity ball to find a conservative estimate of the IF violation. They can then use this bound to certify distributional IF around the empirical data distribution and apply finite sample guarantees to give an estimate of the true distributional IF.

The authors then show how to use the bounds on distributional fairness as regularizers in the training procedure as a way to debias neural networks. They then provide experimental evaluation on a few benchmark datasets that demonstrates that their proposed training method indeed improves distributional individual fairness, at relatively modest degradations in accuracy.

**Strengths:**

The main advantage is a relatively lightweight way to certify and train NNs for IF, in a way that requires little additional computation, compared to previous methods which are not able to scale to large NNs.

The experimental evaluation seems to confirm that DIF training as proposed by the regularization method does in fact improve significantly improve IF at modest degradation in classification accuracy.

**Weaknesses:**

Section 5 is a little dense and it would be helpful for the reader if there was a little more discussion of the optimization procedure, particularly in Section 5.3. Theorem statements here might also be helpful for the reader to understand what the final guarantees are.

**Questions:**

What is the purpose of Table 2? It is a little difficult to interpret the punchline - it just seems to indicate that DIF training does not have a consistent effect on group fairness measures, either positively or negatively.

**Limitations:**

-

---

> ### Author Rebuttal · Authors · 2023-08-09
>
> We thank the reviewer for their valuable comments on the presentation of our paper. Below we comment on the points and questions raised by the reviewer by providing specific actions that will be taken to address these presentation points.
>
> ### On Clarity of Section 5
> We thank the reviewer for this valuable comment. First, we comment that the requested theorem statement already exists in the paper as Theorem E.2 in the Appendix. We agree that moving Theorem E.2 to the main text of the paper and writing it to emphasize the final guarantee will improve the paper, thus, we propose to change the text of Theorem E.2 to the following and to present it in the main text as follows:
>
> Theorem 5.3: A solution to Equation (5), $\bar{\epsilon}$, is a sound upper-bound on the Distributional Individual Fairness violation, and therefore, is a certificate that no $\gamma$-Wasserstien distribution shift can cause the individual fairness of the model $f^{\theta}$ to exceed $\bar{\epsilon}$.
>
> In addition, we will provide a clear statement of Lemma 5.3 (see our comment to Reviewer Lpfq) which adds clarity to the exact guarantees offered by our framework.
>
>
> ### On the purpose of Table 2
>
> The intended purpose of Table 2 is two fold. Firstly, to demonstrate empirically that optimizing a model for distributional individual fairness (DIF) does not inherently worsen nor improve other popular notions of fairness. Secondly, by showing that it has no strong correlation with other notions of fairness we hoped to convey that while DIF is a flexible and powerful notion of fairness, other aspects of fairness should be analyzed to get a wholistic picture of model bias.
>
> To improve the presentation of the paper, we will highlight this point in our Broader Impacts section and move the table and discussion to the Appendix where we can further elaborate on the discussion of this point.

---

### Author Response · Authors · 2023-08-09
**Global Response to Reviewers**

We would like to thank all of the reviewers for their detailed feedback and positive sentiments. The main points for improvement of the paper noted by reviewers were in the presentation of the work. While we have responded to each reviewer detailing how we intend to address their specific points, we list the key modifications of the final paper that will take place to improve presentation.

### Key modifications:

* Moving Section 6.4 and Table 2 to the Appendix while adding the key point they make (that no one fairness metric captures a complete picture of bias) to our Broader Impact statement.
* Add additional theorem statements (see Theorem 5.3 and Lemma 5.3 in response to Reviewer 1 and 2) in order to clarify our theoretical contribution.
* Add in a paragraph clearly explaining the computational complexity advantage of our approach to IF certification versus MILP approaches (see response to Reviwer 2).
* Further clarify and emphasize the novelty and impact of our optimization formulation and the solution approach.

### Key contributions and observations
While we address each of the detailed concerns below, we first state that in the final version of the work we will revise the bullet points at the end of the introduction and add clear text highlighting the following novel contributions:
* An orthotope bounding procedure for Mahalanobis metrics in the context of adversarial robustness/individual fairness.
* A novel optimization formulation to compute upper and lower bounds on the violation of Distributional Individual Fairness property
* A formal proof that shows that our optimization formalization for the upper-bound admits an efficient solution for any differentiable model by leveraging quasi-convex optimization
* We validated our claims through extensive evaluation under real-world distribution shift

We hope that our detailed responses will clear up any reviewer doubts and we are more than happy to answer any further questions the reviewers may have.

---

### Decision · Program_Chairs · 2023-09-21

**Decision:**

Accept (poster)

**Comment:**

The paper proposes a method for certifying individual fairness of neural networks under distribution shifts. Particularly, it seeks to certify that the output of the neural network remains almost the same (for each individual data point) when the input distribution changes within a certain Wasserstein ball. The reviewers found the results interesting for the community. There were several initial concerns about the presentation of the results. However, after a few communication rounds between the authors and the reviewers, the concerns have been resolved. Therefore, I believe the paper can be published after minor changes. Please consider the reviewers' comments in your revision.